# Development of Comprehensive Serological Techniques for Sensitive, Quantitative and Rapid Detection of *Soybean mosaic virus*

**DOI:** 10.3390/ijms23169457

**Published:** 2022-08-21

**Authors:** Rui Ren, Tao Wang, Le Gao, Puwen Song, Yunhua Yang, Haijian Zhi, Kai Li

**Affiliations:** 1MARA National Center for Soybean Improvement, Nanjing Agricultural University, Weigang 1, Nanjing 210095, China; 2College of Agronomy, Henan Agricultural University, Zhengzhou 450046, China; 3Handan Academy of Agricultural Science, Handan 056001, China; 4Department of Horticulture, Beijing Vocational College of Agriculture, Beijing 102442, China; 5College of Life Science and Technology, Henan Institute of Science and Technology, Xinxiang 453003, China

**Keywords:** *Soybean mosaic virus*, SMV-CP, PAb-SMV-CP, MAb-SMV-CP, DAS-qELISA, SMV-GICS

## Abstract

Soybean is an important grain and oil crop worldwide; however, the yield and seed quality of which are seriously affected by *Soybean mosaic virus* (SMV). As efficient detection technology is crucial for the field management of SMV, novel immunological detection methods were developed in the present study. According to the phylogenetic analysis, the CP coding sequence of SMV-SC7 was selected for the prokaryotic expression of the recombinant SMV-CP. Purified SMV-CP was used for the development of polyclonal antibodies (PAb) against the SMV-CP (PAb-SMV-CP) and monoclonal antibodies (MAb) against SMV-CP (MAb-SMV-CP). Subsequently, the PAb-SMV-CP was used for the development of a novel DAS- quantitative ELISA (DAS-qELISA) kit, of which the sensitivity was greater than 1:4000, and this could be used for the quantitative detection of SMV in China. Meanwhile, the MAb-SMV-CP was labeled with colloidal gold, and then was used for the development of the SMV-specific gold immunochromatography strip (SMV-GICS). The SMV-GICS gives accurate detection results through observed control lines and test lines in 5 to 10 min, sharing the same sensitivity as RT-PCR, and can be used for rapid, accurate and high-throughput field SMV detection. The DAS-qELISA kit and the SMV-GICA strip developed in this study are SMV-specific, sensitive, cheap and easy to use. These products will be conducive to the timely, efficient SMV epidemiology and detection in major soybean-producing regions in China and abroad.

## 1. Introduction

Soybean (*Glycine max* (L.) Merr) is an important crop providing oils and protein for human consumption, animal feed production and biofuel all over the world [1]. However, *Soybean mosaic virus* (SMV; *Potyvirus*), as one of the most prevalent viruses, can lead to serious seed quality deterioration and great yield losses (as high as 86%) of soybean under favorable conditions [2,3,4,5,6,7]. SMV, which is seed-borne and aphid-transmitted, usually causes soybean mosaic on leaves, local and systemic necrosis and plant dwarfing [8,9,10]. Highly diverse symptoms in soybean plants infected with SMV resulted in difficulties in the accurate recognition and early removal of the viral pathogen.

SMV was first reported in America, then in Korea, Germany and China at the beginning of the twentieth century [11,12,13,14]. Based upon their reactions in diverse soybean genotypes [15], SMV isolates were classified into seven distinct pathotypes (G1–G7) in the United States and Korea [16,17], while twenty-two SMV strains (SC1-SC22) have been documented in China [18,19,20,21,22,23,24]. Out of these, four of the most prevalent SMV strains, SC3, SC7, SC15 and SC18, have been reported to be widely distributed in the three major soybean-producing regions in China, including Northeast China, Huang-Huai Valleys and Southern China [20,21,22,23,24,25] (Figure 1). The virulence of these SMV strains differs from each other. The virulent strain SC15 broke the resistance of all ten different hosts [21,22,23,24,25,26], and resistance-breaking isolates have been also identified in South Korea, Canada and Iran [27,28,29]. The widespread occurrence of virulent strains poses a serious threat to soybean production. Therefore, the development of sensitive, rapid, low-cost, and high-throughput detection technologies is crucial for the field management of SMV and soybean breeding of SMV-resistant cultivars.

The double antibody sandwich–enzyme-linked immunosorbent assay (DAS-ELISA) is well known as a common immunological technique for virus detection and has now been widely used to detect many plant viruses, such as *Potato virus S* [30], *Citrus yellow vein clearing virus* (CYVCV) [31] and *Zucchini yellow mosaic virus* [32]. Commercial SMV-specific DAS-ELISA kits are available for laboratory use [9,23,33]. However, the coating antibody and the detecting conjugate of the imported DAS-ELISA kits were developed specific to the abroad SMV strains (G1-G7). Hence, it is required to develop a proprietary, low-cost, efficient and substitutable SMV-specific DAS-ELISA system for China. In addition to DAS-ELISA, several detection techniques, such as reverse transcription–polymerase chain reaction (RT-PCR) [33], quantitative RT-PCR (qRT-PCR) [34], reverse transcription loop-mediated isothermal amplification (RT-LAMP) [35,36,37,38], and high-throughput sequencing [39], can be also used to detect SMV in soybean. However, these methods are time-consuming, laborious and expensive, and they require specialized laboratory equipment, which means they are not suitable for large-scale field surveys. The colloidal gold-based immune chromatographic strip (GICS) is currently the quickest technique for plant virus detection. This assay involves antigen–antibody-specific binding, colloidal gold labeling and immunochromatography [40]. GICSs have been used in the diagnosis of plant viral diseases, such as *Citrus tristeza virus* (CTV) [41], *Satsuma dwarf virus* (SDV) [42], *Tobacco mosaic virus* (TMV) [43], *Plum pox virus* (PPV) [40,44], *Lily symptomless virus* (LSV) [45], and *Lily mottle virus* (LMoV) [46], SMV [47], CYVCV [48], *Citrus tristeza virus* (CTV) [49], *Tomato zonate spot tospovirus* (TZSV) [50], *Rice stripe virus* (RSV) [51], *Sugarcane mosaic virus* (SCMV) [52], *Banana bract mosaic virus* (BBrMV) [53] and *Tomato spotted wilt virus* (TSWV) [54]. However, there is no systematical report on the applicability of GICSs to the diagnosis of SMV infection.

Therefore, comprehensive serological techniques for the sensitive, quantitative and rapid detection of *Soybean mosaic virus* were developed in the present study. Purified SMV-CP recombinant protein was obtained and was used for the development of SMV-CP-specific polyclonal antibodies (PAb) and monoclonal antibodies (MAbs). Subsequently, the selected PAb and MAb were used for the development of a novel, sensitive and easy-to-use DAS-qELISA kit and SMV-specific GICS (SMV-GICS). These comprehensive detection techniques would aid in the prevention and control of SMV infection in major soybean-producing regions in China and abroad.

## 2. Results

### 2.1. Biological Purification and Identification of SMV Strains

To obtain purified isolates of SMV strains SC3, SC7, SC15 and SC18, the biological purification and identification of the four SMV strains was carried out. Leaves with typical mosaic or necrosis symptoms were collected from soybean plants infected with the four SMV strains (Figure 2A) and then confirmed to be SMV-positive using the commercial DAS-ELISA kit. Subsequently, the biological purification of the four SMV isolates was conducted as Li et al. described [23] (Figure 2B). Briefly, these SMV strains were back inoculated to the susceptible soybean cv. Nannong 1138-2 (NN 1138-2). Seven to ten days later, soybean plants infected with the four SMV strains showed mosaic symptoms on the first trifoliate leaves. For biological purification, mosaic leaves of NN 1138-2 were collected and inoculated to one-half of a detached, fully expanded *P. vulgaris* cv. Topcrop leaf with a paintbrush. After 2 to 3 days’ incubation, single obvious vein necrosis or local necrotic spots were cut from the leaves and back inoculated to NN1138-2 plants. Each necrotic spot was considered as a purified isolate, and the SMV-positive isolates were verified via DAS-ELISA. Ultimately, purified SMV isolates were inoculated to the ten differential hosts, and homologous isolates were identified in accordance with the reaction of SC3, SC7, SC15 and SC18.

### 2.2. Recombinant SMV-CP Protein

To analyze the intraspecific relationships of the four SMV strains with other identified SMV isolates and potyviruses, sequence alignments and phylogenetic analysis were conducted using the full-length deduced amino acid sequences of SMV-CP (Appendix A). The full-length coding sequence (CDS) of SMV-CP of the four SMV isolates, SC3, SC7, SC15 and SC18, were amplified and sequenced. The *cp* gene of SMV isolates were all 795 bp (encoding 265 amino acids) and were consistent with the reported sequences (Figure 3A). Sequence alignments revealed that SMV-CP was highly conserved between these isolates, which shared 94.72% to 100.00% identity in amino acid sequences (Appendix A). For instance, isolates SC7 (4278-1), SC6-N and KY share the same sequence, while SC7 (4278-1) shares 98.87%, 97.36% and 99.25% identity with SC3, SC15 (6067-1) and SC18 (4424), respectively (Figure 3B, Appendix A). The neighbor-joining tree of SMV-CP resulted in two distinct groups (groups Ⅰ and Ⅱ), and for the four SMV strains, SC3, SC7 and SC18 were clustered in ‘group Ⅰ’, while the virulent strain SC15 was classified to a distinct branch (Figure 3C). Mild strains SC3 and SC18 showed a higher homology, and they were clustered in ‘Subgroup A’, while SC7 was clustered in ‘Subgroup B’. SC7 has the widest distribution in the four SMV strains (Figure 1); hence, the SMV-*cp* gene of SC7 was selected for the expression of the SMV-CP protein. To obtain sufficient SMV-CP antigens for immunization, the full-length CDS of the SMV-*cp* gene was cloned into the prokaryotic expression vector pCZN1 (Appendix A). The recombinant vector pCZN1-SMV-cp was transformed into *Escherichia coli* expression strain Arctic Express BL21 (DE3). The SDS-PAGE showed that the 6× His-tagged recombinant protein SMV-CP (~31.4-KD) was successfully expressed following isopropyl-beta-D-thiogalactopyranoside (IPTG) induction (Figure 3D). Then, the fusion protein was purified with the Ni-NTA resin and collected in the elution buffer containing 250 mM imidazole (Figure 3E). Finally, more than 6.0 mg (1.0 mg/mL in a 1 mL volume) of the recombinant SMV-CP protein was obtained for immunization. 

### 2.3. Preparation and Characterization of SMV-Specific PAb and PAb-HRP

Polyclonal antibodies were raised in rabbit against the purified SMV-CP protein fused with 6×His. Blood samples from the auricular veins of the rabbits were collected for titer evaluation assay using the indirect ELISA method at 35 days post-immunization (DPI). Once the titer of antiserum against SMV-CP protein was greater than 1:50,000, the blood samples were taken to prepare the antiserum. Total blood was collected and centrifugated, and eventually, 1.78 mg (0.89 mg/mL in a 2 mL volume) of the purified antibody was obtained. The titer of the purified PAb-SMV-CP was greater than 1:512,000, which was detected using indirect ELISA (Figure 4A). SDS-PAGE and Coomassie blue staining showed that the purity of the purified antibody was greater than 90% (Appendix A). The purified antibody PAb-SMV-CP was labeled with horseradish peroxidase (HRP), and the indirect ELISA showed that the titer of PAb-SMV-CP-HRP was greater than 1:512,000 (Figure 4B). The PAb-SMV-CP and PAb-SMV-CP-HRP were determined for the applicability of the development of DAS-ELISA. Checkerboard analysis of serial dilutions of capture and detection antibodies showed that the optimal concentration for capture via PAb-SMV-CP was 5 μg/mL, and for detection via PAb-SMV-CP-HRP, it was 0.25 μg/mL. The results showed that the titer of PAb-SMV-CP-HRP was greater than 1:4000, which made it suitable for the development of the DAS-ELISA kit (Figure 4C). The specificity of the DAS-ELISA kit was further confirmed using reactions with the crude extracts from soybean leaves infected with the four SMV strains (SC3, SC7, SC15 and SC18) and the bean common mosaic virus (BCMV) and watermelon mosaic virus (WMV). It was shown that the DAS-ELISA in this study could only be used to detect SMV-CP proteins and SMV-infected leaves but not BCMV and WMV, which indicated that the DAS-ELISA kit was highly specific for SMV (Figure 4D).

### 2.4. Development and Application of the DAS-qELISA Kit

To detect the quantity of SMV in plant tissues with the DAS-qELISA, standard curve was established by using two-fold serially diluted concentration of standard SMV-CP protein. The standard curve between the OD_450_ value and the concentration of the SMV-CP protein was obtained as follows: y = 0.0026x + 0.3082, *R*^2^ = 0.9789, and the minimum detection limit of the SMV-CP protein was approximately 5.0 ng/mL (Figure 5A). To determine the reliability of the DAS-qELISA kit, typical mosaic leaves of soybean plants infected with SC3, SC7, SC15 and SC18, also with the SMV-free control, were collected for quantitative detection of SMV-CP (Figure 5B). According to the pre-experiment results, general extraction of SMV-infected leaves was ten-fold serially diluted (10^−1^, 10^−2^, 10^−3^ and 10^−4^) for accurate detection. Along with the adding of the stop solution, yellow coloration was observed in the DAS-ELISA wells confirming SMV presence (Figure 5C). The OD_450nm_ absorbance values of the three replications of respective samples were measured, and the quantity of SMV-CP were calculated according to the standard curve. Data showed that the SMV-CP present in the mosaic leaves was determined between 300–500 ng/g, and the virulent strain SC7 showed the highest concentration (Figure 5D).

### 2.5. Preparation and Identification of MAbs against SMV-CP

To produce specific MAbs against SMV-CP, six BALB/c female mice were immunized with the purified recombinant SMV-CP protein. Seven days after the fourth immunization, antiserum samples were obtained from the tail vein of each mouse, and MAb presence was determined by an indirect ELISA assay. One of the six mice showed the highest titer (1:121,500) against the recombinant SMV-CP protein (Figure 6A). The mouse SMV-CP-6 that showed the highest titer of the antiserum was selected to prepare the spleen lymphocytes which were then fused with SP2/0 myeloma cells. Screening was performed 10 days after fusion, and 12 positive clones were selected out by indirect ELISA with the supernatants of clones after one week to ten days’ culture (Appendix A). With the secondary screening, 8 out of the 12 positive clones were ultimately selected by indirect ELISA. The eight selected positive hybridoma cell lines were subsequently sub-cloned by injecting into the BALB/c mice. The ascites fluid of the 8 sub-cloned BALB/c mice was collected, and the immunoglobulin class and subclass of the six hybridoma lines (3B6, 5N9, 4D2, 7G1, 4F6, 9H7) were determined as IgG1 (Figure 6B). Ultimately, the best SMV-CP specific BALB/c hybridoma line 9H7 (BALB/c-SMV-CP-9H7) was chosen for the preparation of SMV-CP specific MAb. The MAb of BALB/c-SMV-CP-9H7 was obtained and purified, and eventually 2.28 mg (1.14 mg/mL in a 2 mL volume) of the purified antibodies were obtained. The titer of the purified MAb of BALB/c-SMV-CP-9H7 was detected up to 128,000 (Figure 6C). To test the specificity of the MAb BALB/c-SMV-CP-9H7, the purified MAb was used as the coating antibody for SMV-CP protein detection in the DAS-ELISA. It was showed that the MAb BALB/c-SMV-CP-9H7 could only be used to detect SMV-CP protein and SMV-infected leaves but not for BCMV and WMV (Figure 6D), which indicating the highly specificity of the MAb BALB/c-SMV-CP-9H7.

### 2.6. Controlled Test and Field Application of the SMV-GICS

The MAb BALB/c-SMV-CP-9H7 labeled with colloidal gold was then used for the SMV-GICS development (Figure 7A). To test the sensitivity of the SMV-GICS, the crude leaf extract of soybean plants infected with SMV-SC7 was serially diluted (10^−1^, 10^−2^, 10^−3^ and 10^−4^), and 100 μL of each dilution was added to the sample pad of the strip. The crude extract of SMV-free soybean leaves was used as the negative control. All of the positive control lines strongly turned purple, while no color development was viewed on the test lines for the SMV-negative control, indicating the availability of SMV-GICS (Figure 7B). All of the test lines for the gradient-diluted, SMV-positive leaf suspension turned purple, and the color became darker and then lighter as the dilution increased (Figure 7B). These results indicated that the SMV-GICS could still give a positive signal with a 1000-fold dilution of the SMV-infected leaf sample of soybean, and it works best when diluted ten times. To determine the specificity of the SMV-GICS, they were tested with the crude extracts of soybean leaves infected with SMV, BCMV and WMV. Positive results were only observed on the strips tested with the SMV-infected samples (SC3, SC7, SC15 and SC18), but those of BCMV and WMV were negative (Figure 7C). These results suggested that because of the high specificity of the SMV-GICS, it can only be used to detect SMV rather than other potyviruses. Moreover, RT-PCR was conducted to verify the sensitivity of the SMV-GICS. Briefly, the cDNA originated from the soybean leaves infected with SC3, SC7, SC15 and SC18 was diluted with gradient concentrations (10^−1^, 10^−2^ and 10^−3^) for the RT-PCR detection of SMV. The target-fragments of the SMV-*cp* were amplified from all of the SMV-positive samples, while no product was amplified from the negative- control and blank control (Figure 7D). This suggested that the SMV-GICS and RT-PCR share the same sensitivity in the detection of SMV.

Additionally, 18 soybean field samples (S1–S18) which appeared the mosaic, shrinkage, necrosis, etc., symptoms, were detected via the SMV-GICS (Figure 8A). Out of the 18 samples, 11 were positive (S1, S2, S3, S4, S5, S6, S7, S9, S10, S11 and S12) and 7 were negative (S8, S13, S14, S15, S16, S17 and S18) (Figure 8B). RT-PCR was conducted to verify the presence of SMV in the 18 samples. The RT-PCR results showed that 12 samples were positive and 6 were negative (Figure 8C). However, RT-PCR showed a slightly higher positive ratio (66.7%) than that of the SMV-GICS (61.1%). Collectively, all these results suggested that the SMV-GICS can be used for the rapid and accurate detection of SMV in the soybean field.

## 3. Discussion

Among the various bacterial, fungi and viral pathogens infecting soybean, SMV is present in soybean-growing areas all over the world. SMV infection can significantly reduce the quantity and quality of soybean seeds (e.g., mottled seed coats, reduced seed size and viability) [8]. SMV can be transmitted from plant to plant by vectors and seeds, making it difficult to prevent their rapid spread [54]. The management of SMV is limited to the use of good agricultural practices and the development of resistant cultivars through breeding and genetic engineering [55]. Early detection is an important component of virial disease management [56,57,58]. Therefore, comprehensive serological techniques including the DAS-qELISA kit and the SMV-GICS were developed for the sensitive, quantitative and rapid detection of SMV infection in soybean plants in the current study (Figure 9). The novel DAS-ELISA kit developed not only enabled the quantitative detection of SMV, but also exhibited a greater sensitivity than that of commercial DAS-ELISA kits. Moreover, the SMV-GICS can be used for rapid and accurate field SMV detection within 5–10 min, sharing the same sensitivity as RT-PCR. The two SMV detection methods established herein would greatly facilitate the early detection and field management of SMV.

The genome of SMV is one single-stranded, positive-sense RNA with a total length of about 9.6 kb nucleotides (nt), containing a single open reading frame (ORF). The single ORF encodes a large polyprotein (about 350 kDa) which is ultimately cleaved and processed to form 11 multifunctional proteins [59,60,61] (Figure 3A). It is reported that helper-component proteinase, P3 and cytoplasmic inclusion are probably the SMV elicitors of gene-mediated resistance [62,63,64,65,66,67,68], and P3 facilitates SMV replication [69]. The SMV-CP, the only structural protein of SMV, plays an important role in aphid transmission [70,71,72]. The SMV-CP sequences of various isolates are highly conserved [73], which agrees with our results, as they shared 94.72% to 100.00% identity in amino acid sequences (Figure 3B, Appendix A). Hence, the SMV-CP was used as the antigen in the DAS-ELISA, as well as the indicator gene in the (q)RT-PCR detection of SMV [33,34]. As the serological detection of a virus is based on the specific binding of viral proteins with antibodies [74,75], the diversity of SMV-CP might lead to the weakened sensitivity and accuracy of the commercial DAS-ELISA kits in detecting SMV samples in China. Based on the sequence alignment and phylogenetic analysis of SMV-CP, the SMV-*cp* gene of the widest distributed SMV-SC7 strain was selected for the expression of SMV-CP proteins and the obtainment of SMV-CP and SMV-CP-PAb (Figure 3C). In addition, PAb and MAbs specific to SMV-CP were further developed for the SMV-DAS-qELISA kit and the SMV-GICS using the purified SMV-SC7-CP recombinant protein. The biological purification and molecular identification of SMV strains provided a substantial basis for the development of highly specific, highly sensitive and widely available SMV-detection tools.

The high cost and low sensitivity of commercial SMV-specific DAS-ELISA kits make their application in high-throughput field SMV detection unrealistic. Therefore, a SMV-specific DAS-ELISA kit was developed using the novel SMV-CP-PAb and SMV-CP-PAb-HPR antibodies obtained. The sensitivity of our DAS-qELISA kit was greater than 1: 4000 (Figure 4C), which was higher than that of the commercial DAS-ELISA kits (1:810–1:2430) (http://www.nanodiaincs.com/smv.htm, accessed on 5 July 2022). Additionally, with the establishment of the standard curve between the OD_450nm_ value and the concentration of the SMV-CP protein (Figure 5A), our DAS-qELISA kit can be used for the quantitative detection of SMV present in various plant tissues. It enables the assessment of virus content and the resistance levels of soybean breeding materials, which provide important support for the quarantine of soybean international trade and for the fine breeding of soybean cultivars resistant to SMV [34].

Compared with the established plant virus detection methods (e.g., RT-PCR, RT-LAMP and RNA sequencing), the GICS has several advantages over traditional serological assays due to its simplicity, speediness and limited requirements for work experience and equipment [75,76], which would allow the high-throughput investigation of viral infections. Various GICSs have been widely applied in the diagnosis of viral diseases infecting citrus (including CTV, SDV, CYVCV and CTV) [41,42,48,49], tobacco (including TMV, TZSV and TSWV) [43,50,54], lily (LSV and LMoV) [45,46], and other horticultural crops (PPV, SCMV and BBrMV) [44,52,53]. However, few GICSs have been developed for virus detection in annual field crops [47,51]. Herein, MAbs against SMV-CP was developed and well-characterized (Figure 6), and then, a novel GICS was developed using the best MAb of BALB/c-SMV-CP-9H7 labeled with colloidal gold (Figure 7). The SMV-GICS would still give a positive signal with a 1000-fold dilution of the SMV-infected leaf sample of soybean, which indicates its high sensitivity (Figure 7B). Meanwhile, we noted that the GICS gave positive results when the crude leaf extract was diluted more than ten times (Figure 7B). It was inferred that impurities and pigments in the crude extract might influence the release of the colloidal gold-BALB/c-SMV-CP-9H7 conjugates and the binding capacity of antibody [49]. The SMV-GICS was further tested to determine its high specificity in detecting SMV rather than other potyviruses (Figure 7C). Moreover, the SMV-GICS was applied for the field detection of SMV, and only 1 out of the 18 samples showed false negative results when compared with RT-PCR detection (Figure 7D). These results indicated similar sensitivities between RT-PCR and the SMV-GICS. In addition, the SMV-GICS would give accurate detection results through observed control lines and test lines in 5 to 10 min. This suggests that SMV-GICS possesses an irreplaceable advantage in the rapid and high-throughput field detection of SMV compared with DAS-ELISA, RT-PCR, RT-LAMP and sequencing.

Taken together, the DAS-qELISA kit and the SMV-GICS developed in this study is SMV-specific, highly sensitive, efficient, low-cost and easy to operate, which should benefit SMV management in China and abroad.

## 4. Materials and Methods

### 4.1. SMV Isolates and Plant Materials

The four SMV strains: SC3, SC7 (isolate 4278-1), SC15 (isolate 6067-1) and SC18 (isolate 4424) were maintained in the susceptible soybean cultivar NN 1138-2. The bean *Phaseolus vulgaris* cv. Topcrop was used for the biological purification of SMV isolates. The soybean NN 1138-2 combined with nine other different resistant soybean cultivars including Youbian 30, 8101, Tiefeng 25, Davis, Buffalo, Zaoshu 18, Kwanggyo, Qihuang No. 1, and Kefeng No. 1 were used for the identification of the four SMV strains. All of the SMV isolates and plant materials used in this study were provided by the National Center for Soybean Improvement, Nanjing Agricultural University, Nanjing, China.

### 4.2. Inoculation, Biological Purification and Identification of SMV Strains

The inoculum was prepared by grinding young symptomatic leaves with a mortar and pestle in 0.1 M sodium phosphate-buffered saline (PBS, pH 7.4) at a ratio of 1:2 (weight/volume (*w*/*v*)) mixed with a small amount of carborundum powder (600-mesh). Fully expanded primary leaves of soybean plants were mechanically inoculated by gently rubbing them with the inoculum using a paintbrush; thereafter, tap water was sprayed onto the inoculated leaves as Li et al. described [23]. The biological purification of the four SMV isolates was performed as Li et al. described [23]. Briefly, symptomatic leaves of NN 1138-2 were collected and inoculated to one-half of a detached, fully expanded leaf of the bean cultivar ‘Topcrop’ with a paintbrush. Afterward, inoculated leaves were immediately rinsed with running tap water. Finally, they were put into Petri dishes containing moist filter paper and incubated in a growth chamber at 25 to 30 °C, 90% humidity and 48 to 72 h of continuous incandescent lighting. After 2 to 3 days, SMV infection symptoms such as a single obvious vein necrosis or local necrotic spots developed. Following infection symptom development, single diseased spots were cut from leaves with sterilized scissors, then extracted into 10 mM PBS and inoculated back onto NN1138-2 plants as outlined above. Each necrotic spot was designated as a purified isolate, and each virus sample was purified by passage through *P. vulgaris* cv. ‘Top crop’. The SMV- positive isolates were verified using a commercial DAS-ELISA kit (Nano Diagnostics, LLC, Fayetteville, AR, UAS) [23]. Ultimately, purified SMV isolates were inoculated to the ten differential hosts, and homologous isolates were identified.

### 4.3. Sequence and Phylogenetic Analysis

Total RNA was extracted from young leaves of Nannong 1138-2 plants infected with the purified SC3, SC7, SC15 and SC18 strains using an RNA Simple Total RNA Kit (Tiangen, Beijing, China). The DNA-free RNA was used for first-strand cDNA synthesis with Oligo (dT) primers and a PrimeScript™ И 1st strand cDNA Synthesis Kit (Takara, Dalian, China) following the manufacturer’s instructions. Using the Primer Premier 5.0 software (Premier, Palo Alto, CA, USA), the primers SMV-*cp*-F and SMV-*cp*-R were designed according to the full-length CDS of the *SMV-cp* gene (Appendix A). Subsequently, the fragments for the SMV-*cp* gene of the four purified SMV strains were amplified with the cDNA using Primer STAR Max^®^ DNA Polymerase (Takara, Dalian, China). Purified PCR products were cloned into the pMD18-T vector (Takara, Dalian, China) and verified via sequencing. The obtained CDSs of the SMV-*cp* gene of the four SMV strains were aligned and analyzed for identity using DNAMAN^TM^ (LynnonBiosoft version 8.0, Pointe-Claire, QC, Canada) software together with other SMV isolates available in GenBank (Appendix A). Sequence alignments and phylogenetic analysis were performed based on the full-length amino acid sequences of SMV-CP of the four SMV strains and relevant potyvirus strains/isolates’ sequences using MEGA 5.10 software (Appendix A). The phylogenetic tree was generated using a bootstrap neighbor-joining tree, and the bootstrap values were calculated using 1000 random replications. Two *potyviruses* including WMV-Fr (isolate code AY437609) and BCMV-Y (AJ312438) were used as outgroups.

### 4.4. Prokaryotic Expression and Purification of SMV-CP Recombinant Protein

To obtain the full-length SMV-cp gene, specific primers, pCZN1-SMV-*cp*-F and pCZN1-SMV-*cp*-R, were designed according to the SMV genome sequence of SC7 strain (Appendix A). The two Nde I and Xba I restriction endonuclease sites were arranged to the 5′-end of the forward and reverse primers, respectively. The full-length SMV-cp gene was amplified from the pMD18-T vector containing the SMV-cp fragment using Primer STAR Max DNA Polymerase (Takara, Dalian, China). The PCR product was purified, digested with Nde I and Xba I endonucleases and then cloned into the prokaryotic expression vector pCZN1 (Appendix A). The recombinant plasmid pCZN1-SMV-*cp* was verified through DNA sequencing and then transferred into the Escherichia coli expression strain Arctic Express BL21 (DE3). The expression and purification of SMV-CP was conducted as previously described [77,78]. Briefly, the *E. coli* cells containing pCZN1-SMV-*cp* were cultured in 5 mL of LB medium (containing 50 μg/mL of ampicillin) at 37 °C for 8 h. Then, 1 mL of the culture was transferred into 1 L of LB medium (containing 50 μg/mL of kanamycin) at 37 °C to reach the optical density of OD_600_ = 0.6. Following IPTG (0.5 μM)-induced expression and Ni-NTA purification, the obtained recombinant SMV-CP fusion protein was measured for the concentration using an ultraviolet spectrophotometer Nano-Drop2000 (Thermo Fisher Scientific, San Jose, CA, USA). The purified recombinant protein was dialyzed into 1 × PBS to remove the imidazole. The expression and purification of SMV-CP was determined with SDS-PAGE and stained with Coomassie brilliant blue.

### 4.5. Indirect ELISA

Flat-bottomed polystyrene microtiter plates (Corning, New York, NY, USA) were coated with 2 μg/mL of SMV-CP and incubated at 4 °C for 8 h. The coated plates were washed once with PBST and blocked with 200 μL of 5% milk in PBST at 37 °C for 1 h. After one wash, 50 μL of antiserum diluted with 5% milk in PBST (1:100–1:320,000, volume/volume (*v*/*v*)) was added to each well of the plates; then, the plates were incubated at 37 °C for 1.5 h with cognate antibodies. After being washed three times, 50 μL of HRP conjugated Goat anti-rabbit IgG antibody (GR-IgG-HRP) (Sigma, St. Louis, MO, USA) (1:30,000 in 5% milk in PBST) was added to each well of the plates and incubated for one hour at 37 °C. The plates were washed three times, and then, 50 μL of the 1 mg mL^−1^ 3,3′, 5,5′-tetramethylbenzidine (TMB; Sigma, St. Louis, MO, USA) was added to each well of the plates. The color development was stopped by adding 0.1 M NaOH (50 μL per well). The absorbance was measured in single wave length mode at 450 nm using a Bio-Rad iMark^TM^ microplate absorbance reader (Bio-Rad Laboratories, Hercules, CA, USA) within 30 min.

### 4.6. Preparation of Horseradish Peroxidase Labeled Polyclonal Antibody

The preparation of PAb was conducted as Wu et al. previously described [79]. Two female New Zealand White rabbits (two months old) were immunized subcutaneously with 400 μg of purified recombinant SMV-CP fusion protein emulsified with 400 μL of Freund’s complete adjuvant at a ratio of 1:1 (*v*/*v*) [80]. Two weeks after the first immunization, the rabbits were boosted with four additional subcutaneous injections with 400 μg of the purified protein mixed with 400 μL of Freund’s incomplete adjuvant per injection at a ratio of 1:1 every week. At 35 DPI, blood samples from the auricular vein of the rabbits were collected for the titer evaluation assay. The titer of the antiserum against SMV-CP proteins was determined using the indirect ELISA method. When the titer was greater than 1:50,000, blood samples were taken to prepare the antiserum. Total blood was collected, and the crude antiserum was obtained via centrifugation. The crude polyclonal antibody was purified with a protein A spin kit (Thermo Fisher Scientific, San Jose, CA, USA) according to the manufacturer’s instructions. The titer of the purified antibody was detected via indirect ELISA, and the concentration of the obtained antibody was determined using a BCA protein concentration determination kit (Thermo Fisher Scientific, San Jose, CA, USA). The purity of the purified antibody was observed via SDS-PAGE and Coomassie blue staining. The resulting PAb-SMV-CP was mixed with glycerol and sodium azide to final concentrations of 50% and 0.1%, respectively. The purified PAb-SMV-CP was conjugated with HRP at a mass ratio of 1:1. The HRP labeling PAb-SMV-CP (PAb-SMV-CP-HRP) was detected for titer via indirect ELISA and then was used for the development of the DAS-qELISA kit.

### 4.7. DAS-qELISA

To obtain the standard curve, gradient concentrations of SMV-CP protein (0, 0.98, 1.95, 3.91, 7.81, 15.63, 31.25, 62.50, 125.00, 250.00, 500.00 and 1000.00 ng mL^−1^) were prepared. Meanwhile, about 200 mg of soybean leaves infected with SC3, SC7, SC15 and SC18 was collected and ground with a mortar and pestle in liquid nitrogen, after which, the tissue powder was homogenized with 350 μL of general extraction buffer. DAS-qELISA was performed as described with specific modifications [81,82,83]. Briefly, the 96-well ELISA-plate was coated with 100 μL of coating antibody (PAb-SMV-CP) diluted in coating buffer (1:1000, volume/volume (*v*/*v*), 5 μg ml^−1^) at 4 °C overnight. Following washing with the PBST for 4–6 times, 100 μL of standard SMV-CP protein solution and the gradient-diluted crude extract of leaves infected with SC3, SC7, SC15 and SC18 were added to the wells of the coated ELISA-plate with three replications. After incubation at 4 °C overnight, the wells were washed 4–6 times using the PBST followed by the addition of PAb-SMV-CP-HRP diluted in enzyme conjugate (1:1000, *v*/*v*) solution and incubated at 37 °C for one hour. The wells were washed 4–6 times using wash buffer followed by the addition of 100 μL of substrate solution containing 1 mg/mL of TMB. The plate was incubated at room temperature for 10–60 min, followed by the addition of 100 μL of stop solution (1 M HCl) into each well. The absorbance values were measured at the 450 nm wave length (OD_450nm_ value) using a Bio-Rad iMark^TM^ microplate absorbance reader (Bio-Rad Laboratories, Hercules, CA, USA). The quantitative detection of SMV-CP proteins was calculated according to the optical density (OD_450nm_) average value of the three replications. Meanwhile, SMV-free soybean leaves were used as the negative control, and extraction buffer was added as the blank control.

### 4.8. Preparation of Monoclonal Antibodies

For SMV-CP-specific MAb preparation, six six-week-old BALB/c mice were purchased from the Shanghai Laboratory Animal Center, Chinese Academy of Sciences (certificate of animal quality: Zhong Ke Dong Guan No. 003). The purified recombinant protein SMV-CP was mixed with an equal volume of Freund’s complete adjuvant with repeated stirring to prepare the water-in-oil emulsion. The emulsion (containing 100 μg of antigen protein per mouse) was injected into the peritoneal cavity of the 6-week-old BALB/c female mice. Two weeks later, the emulsion (containing 50 μg of antigen protein per mouse) was injected into the mice every 14 days; this operation was repeated four times. Seven days after the last immunization, antiserum samples were obtained from the tail vein of each mouse. The antiserum was tested for the titers and specificity to the virus via indirect ELISA. The titers of the antisera from the five mice were determined by measuring the binding of serial dilutions of the antisera to the SMV-CP protein using indirect ELISA. The mouse that showed the highest titer of the antiserum was selected to prepare the lymphocytes from the spleen. SP2/0 myeloma cells were cultured in high-glucose DMEM added with 20% fetal bovine serum. The spleen lymphocytes were fused with SP2/0 myeloma cells at a ratio of 20:1 under the agent of the PEG4000 (50%, *w*/*v*) [84]. The fused cells were distributed in the 96-well culture plates at an approximate density of 4 × 10^4^ cells/μL of HAT per well. The selected positive hybridoma cell lines were subsequently sub-cloned via the limiting dilution method. The hybridoma cell lines were injected into the BALB/c mice via the intraperitoneal injection of 2 × 10^7^ hybridoma cells. The ascites fluid was collected 7 to 10 days after the injection. The immunoglobulin subclass and light chain isotype of the antibodies were determined using a Mouse Monoclonal Antibody Isotyping kit (Sigma, St. Louis, MO, USA). The MAb was purified with a protein A spin kit (Thermo Fisher Scientific, San Jose, CA, USA). The titer of the purified Mab (BALB/c-SMV-CP-9H7) was detected via indirect ELISA, and the concentration of the MAb was determined using the BCA protein concentration determination kit.

### 4.9. Preparation of Colloidal Gold-Conjugated Anti-SMV-CP MAbs

Colloidal gold particles (20 nm in diameter) were prepared as described by Zhang et al. [45,46]. A 100 mL solution of 0.01% gold chloride was boiled to reflux station, and then, 1.1 mL of 1% tri-sodium citrate solution was added rapidly with constant stirring. The solution was boiled for additional 5 min until the color of the mixture changed to a brilliant wine red. After being cooled down to room temperature, the gold colloidal solution was added with 0.2 M K_2_CO_3_ to bring the pH to 8.0, and then, the solution was stored at 4 °C. For colloidal gold labeling, 1 mL of Milli-Q purified water with 1000 μg of purified MAb was added drop by drop into 100 mL of colloidal gold solution (pH 8.2) and the mixture was stirred gently, and then, the solution was incubated at room temperature for about 1 h. A total of 10 mL of 10% (0.10 g/mL) bovine serum albumin (BSA) (Sigma, St. Louis, MO, USA) solution containing 0.01 M sodium borate was added slowly to the colloidal gold/Mab solution until it reached the final concentration of 1% (0.01 g/mL). The obtained loose sediment of gold-labeled mAb was then resuspended with 5 mL of 10 mM Tris-HCl buffer (pH 7.4), with 0.01 M sodium borate, 3% (0.03 g/mL) cane sugar, 3% (0.03 g/mL) BSA and 0.05% (0.5 g/L) sodium azide, and the final product was stored at 4 °C.

### 4.10. The Development and Test Procedure of the Immunochromatographic Strips

An immunochromatographic strip was composed of a sample pad, a conjugate pad, a nitrocellulose (NC) membrane and an absorbent pad. Polyester membranes, absorbent papers, sticky bases and plastic cases were purchased from Shanghai Jieyi Biotechnology, China. Conjugate pads (glass fiber membranes) were purchased from Millipore (Bedford, MA, USA) and soaked in 20 mM phosphate buffer (pH 7.4) containing 2% (0.02 g/mL) BSA, 2% (0.02 g/mL) sucrose and 0.1% (1.0 g/L) sodium azide, followed by 24 h of drying in a 37 °C incubator. Colloidal gold immunochromatographic strips were made using the method described by Zhang et al. [45,46]. Briefly, under optimal conditions, the anti-SMV MAb (0.33 mg/mL) and goat anti-mouse antibodies (0.5 mg/mL) were dispensed to the membrane at the test (T) and control (C) lines using the Quanti 3000 BioJets attached to a BioDot XYZ-3000 dispensing platform (Bio-Dot, CA, USA), respectively. The assembled plate was cut longitudinally with a guillotine cutter to produce strips (60 mm × 3 mm) and was packaged inside plastic containers and then stored at room temperature under dry conditions. About 50 mg of soybean leaf tissue was ground into homogenate in 0.1 M sodium PBS at a ratio of 1:2 (*w*/*v*) in a 1.5 mL centrifuge tube with a plastic abrasive rod. Subsequently, the leaf suspension was diluted with gradient concentrations (10, 100 and 1000 times). The diluted mixture was loaded onto the sample pad of the horizontally laid strip. After 5 to 10 min of incubation at room temperature, samples showing two red lines were considered SMV-positive, while those showing only one red control line were SMV-negative. A strip without the control line was considered to be invalid.

### 4.11. Detection of SMV Infection through RT-PCR

The presence of SMV in soybean plants infected with the four strains was examined via RT-PCR. Total RNA was extracted from the leaves with typical mosaic symptoms of soybean plants infected with SC3, SC7, SC15 and SC18 using an RNA Simple Total RNA Kit (Tiangen, Beijing, China). The first-strand cDNA was obtained from the total RNA with Oligo (dT) primers and a PrimeScript™ И 1st strand cDNA Synthesis Kit (Takara, Dalian, China). The cDNA was diluted with gradient concentrations of 10 and 100 times for the sensitivity analysis of the RT-PCR detection of SMV. The primers SMV-*cp*-RT-F and SMV-*cp*-RT-R, specific for the SMV-*cp* gene, and the primers Q-*GmTubulin*-F and Q-*GmTubulin*-R, specific for the soybean *GmTubulin* gene as controls, were synthetized (Appendix A). RT-PCR was performed in a 20 μL volume comprising 2.0 μL of first-strand cDNA, 0.8 μL of each forward and reverse primer (10.0 μM), 10.0 μL of 2 × Taq Master Mix (Vazyme Biotech Co., Ltd., Nanjing, China) and 6.4 μL of sterile distilled H_2_O. All reactions were performed in 200 μL centrifuge tubes using a Bio-Rad T100™ Thermal Cycler (Bio-Rad, Hercules, CA, USA). Finally, PCR products were evaluated using 2% agarose gel electrophoresis.

### 4.12. Statistical Analysis

The statistical analysis was performed with SPSS Version 18.0 software (SPSS Inc., Chicago, IL, USA). All experiments were replicated three times. Data are presented as mean and standard error from three independent experiments. Significant differences among treatments were determined at *p* ≤ 0.05 and *p* ≤ 0.01 based on the least significant difference test via the *Mann–Whitney U*-test.

## Figures and Tables

**Figure 1 ijms-23-09457-f001:**
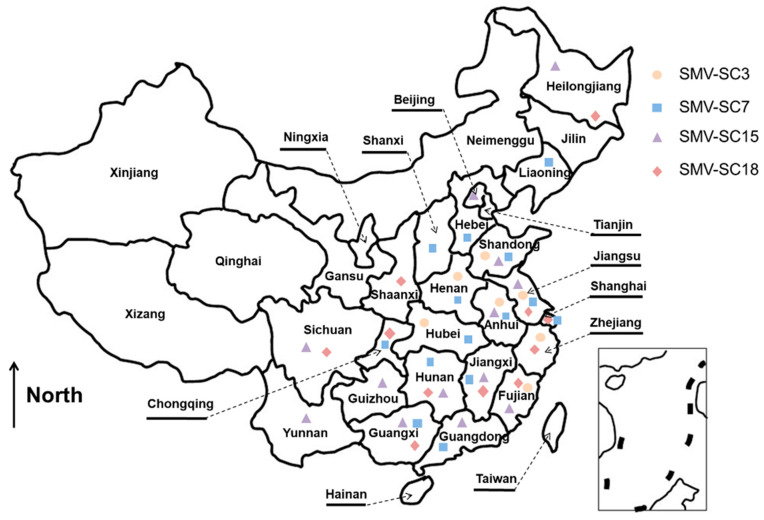
The geographical distribution of the four prevalent SMV strains, SC3, SC7, SC15 and SC18, in China. The orange dots, light-blue squares, purple triangle and brownish-red diamond indicate SMV strains collected in corresponding regions.

**Figure 2 ijms-23-09457-f002:**
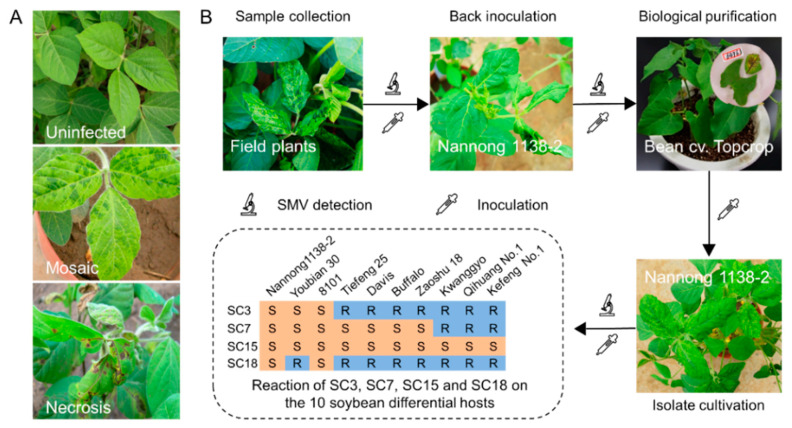
Biological purification and identification of the four SMV strains. (**A**) The typical mosaic and necrosis symptoms caused by SMV infection on soybean plants; (**B**) workflow diagram of the biological purification and identification of the four SMV strains.

**Figure 3 ijms-23-09457-f003:**
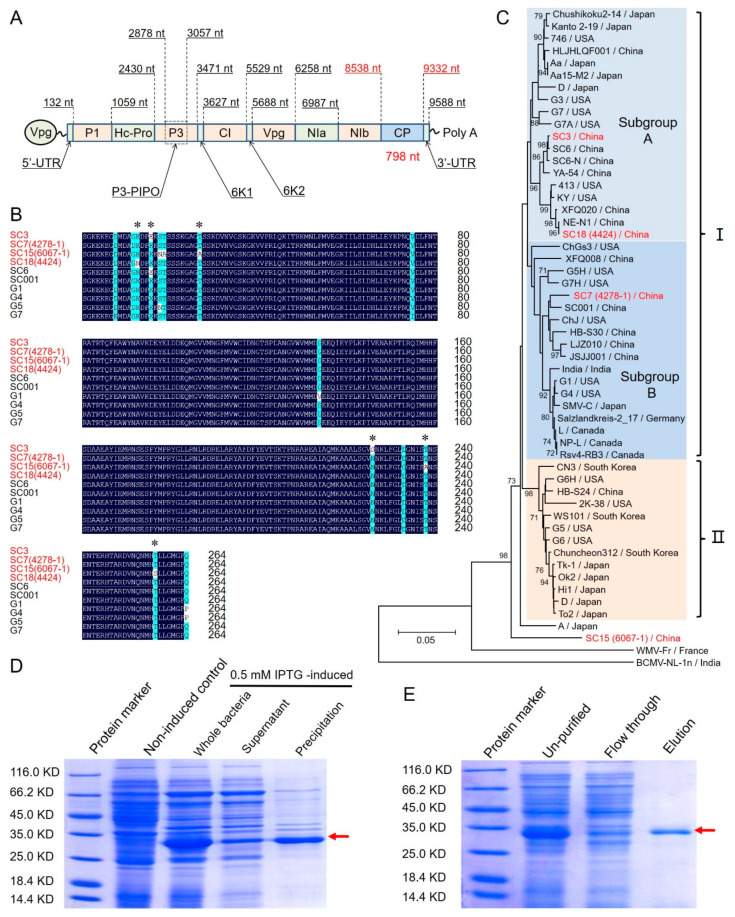
Cloning, sequence analysis and prokaryotic expression of the SMV-cp gene. (**A**) The genome organization of SMV; (**B**) sequence alignment of the SMV-cp genes of SC3, SC7, SC15 and SC18 with other prevalent SMV strains in China (SC6 and SC001) and America (G1, G4, G5 and G7); (**C**) the phylogenetic tree of SMV isolates structured based on deduced SMV-CP amino acid sequences. The phylogenetic tree was constructed using the neighbor-joining method with the 1000 bootstrap values indicated; (**D**) the expression of SMV-CP was determined via sodium dodecyl sulfate–polyacrylamide gel (SDS-PAGE) analysis; (**E**) the purification of SMV-CP was also determined via SDS-PAGE. * indicates polymorphic site in amino acid sequences of SC3, SC7, SC15 and SC18 in Figure 3B; Different colors gray, light blue and orange in Figure 3B indicate different groups, and I and II indicate subgroups in the phylogenetic tree; Red arrows indicate the recombinant protein SMV-CP.

**Figure 4 ijms-23-09457-f004:**
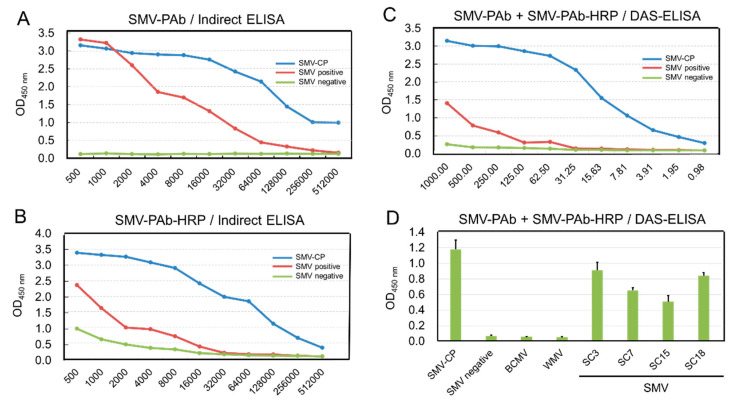
Development and characterization of SMV-CP-specific PAb. (**A**) The titer of PAb-SMV-CP against SMV-CP recombinant protein determined by indirect ELISA; (**B**) The titer of PAb-SMV-CP-HRP detected by indirect ELISA; (**C**) The DAS-ELISA of SMV-CP protein by DAS-ELISA; (**D**) Specificity tests of the DAS-ELISA for SMV detection.

**Figure 5 ijms-23-09457-f005:**
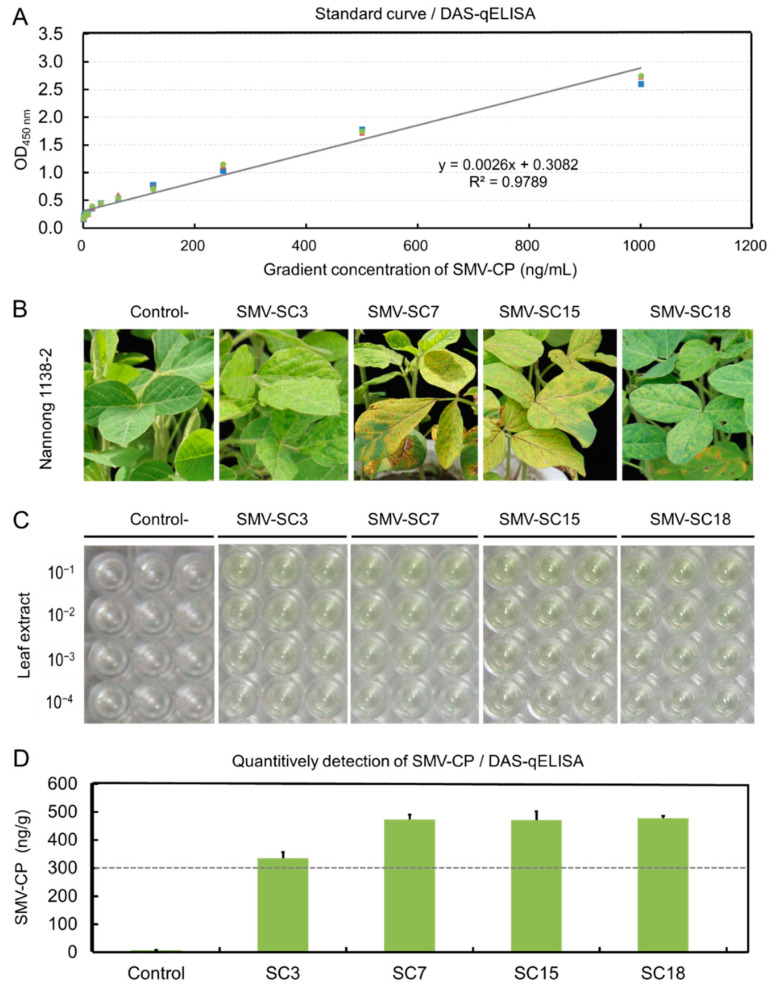
Development and characterization of the DAS-qELISA. (**A**) Standard curve establishment of DAS-qELISA kit for quantitative detection of SMV-CP protein; (**B**) Symptoms of the susceptible genotype NN1138-2 to the for SMV strains SC3, SC7, SC15 and SC18; (**C**) The abundance of SMV in the gradient-diluted crude extract of leaves infected with SC3, SC7, SC15 and SC18 was determined by DAS-ELISA with three replicates; (**D**) Quantitative detection of SMV-CP protein in soybean leaves infected with SC3, SC7, SC15 and SC18. Colored symbols in the subfigure (**A**) indicate different experimental groups.

**Figure 6 ijms-23-09457-f006:**
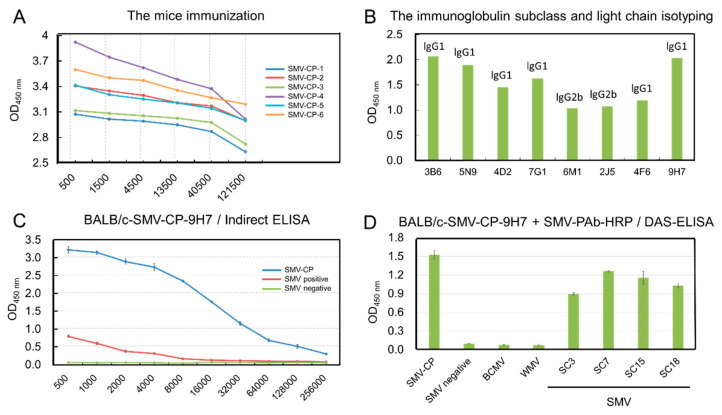
Development and characterization of SMV-CP specific MAb. (**A**) The titer of antiserum against SMV-CP recombinant protein was determined via indirect ELISA; (**B**) the immunoglobulin subclass and light chain isotyping of the SMV-CP-specific MAbs; (**C**) the titer of the purified Mab BALB/c-SMV-CP-9H7 was detected via indirect ELISA; (**D**) specificity tests of the BALB/c-SMV-CP-9H7 for SMV detection was determined via DAS-ELISA.

**Figure 7 ijms-23-09457-f007:**
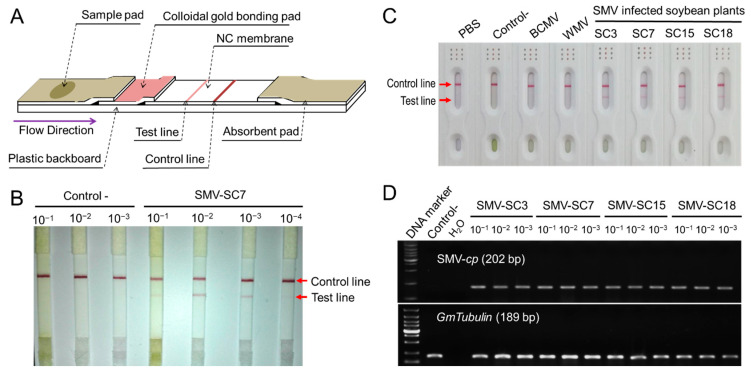
Development, sensitivity and specificity test, and field application of the SMV-GICS. (**A**) The schematic representation of the SMV-GICS; (**B**) sensitivity test of the SMV-GICS in detection of SMV present in the gradient-diluted crude extract of leaves infected with SC7; (**C**) specificity test of the SMV-GICS in detection of SMV, BCMV and WMV; (**D**) detection of SMV present in the leaves infected with SC3, SC7, SC15 and SC18 using RT-PCR.

**Figure 8 ijms-23-09457-f008:**
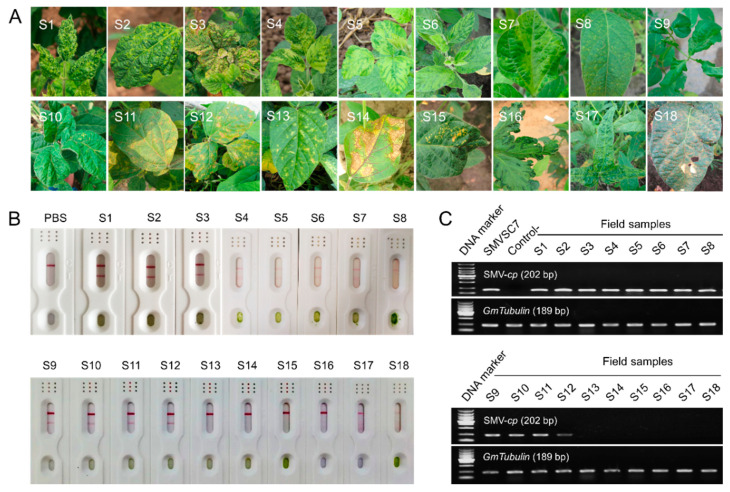
Field application of the SMV-GICS. (**A**) SMV-infected soybean plants in the field of the Huang-Huai-Hai main soybean production region in China; (**B**) detection of SMV in field samples using the SMV-GICA strip; (**C**) detection of SMV present in the leaves of the field plants using RT-PCR.

**Figure 9 ijms-23-09457-f009:**
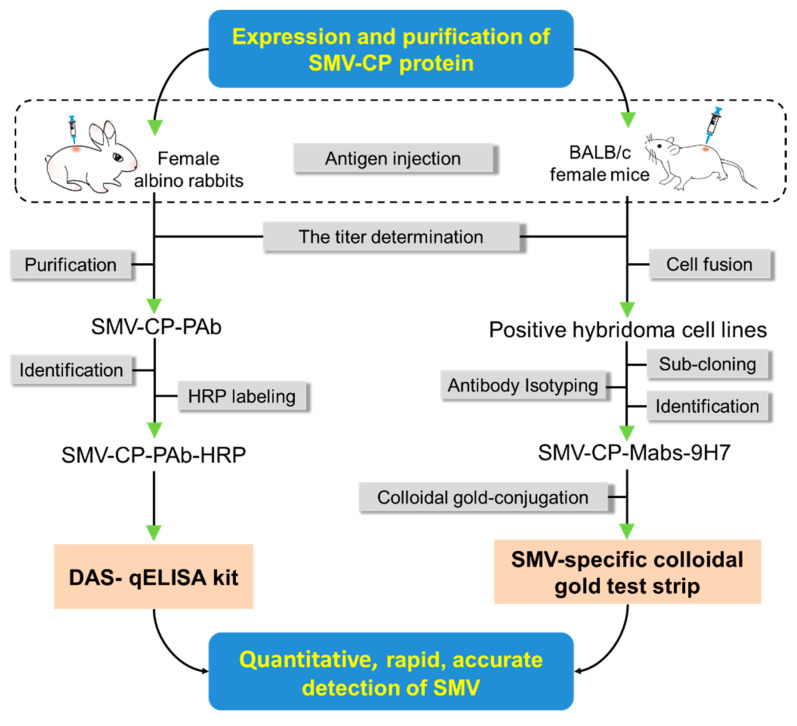
The workflow diagram for development of DAS-qELISA kit and SMV-GICA strips.

## Data Availability

All data are shown in the main manuscript and in the Appendix A.

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
