# Peer review of "Development of Comprehensive Serological Techniques for Sensitive, Quantitative and Rapid Detection of Soybean mosaic virus"

_ijms, 2022, doi:10.3390/ijms23169457_

Round 1

Reviewer 1 Report

The manuscript of Ren et al. describes the development and field application of ELISA and GIC specific to SMV coat protein. Results from this study have comprehensive applications including but not limited to virus detection in the field, breeding, and quarantine. Methods and results are well described and the manuscript is well written. Based on my comments, the manuscript needs only minor edits.

Comments are in the attached Pdf.  

Author Response

Comment 1: The manuscript of Ren et al. describes the development and field application of ELISA and GIC specific to SMV coat protein. Results from this study have comprehensive applications including but not limited to virus detection in the field, breeding, and quarantine. Methods and results are well described and the manuscript is well written. Based on my comments, the manuscript needs only minor edits.

Answer: We have addressed all the issues and revised the manuscript carefully. The changes made in the manuscript are tracked and the itemized list of responses to the comments is as following.

Issues to be addressed:

Comment 2: What is the basis of saying no-significant correlation between SMV population diversity and geographical isolation? Did you conducted population genetic analysis?

Answer: As shown in the neighbor-joining tree of SMV-CP (Figure 3C), SMV isolates all over the world were grouped in two distinct groups (groups â…  and â…¡). Some studies have shown that phylogenetic analysis revealing a significant geographical association among SMV strains/isolate (Seo et al. 2009; Gao et al. 2022). However, SMV isolates from different countries (such as China, Japan, Korea and America) seemed were clustered randomly rather than geographical grouped in the neighbor-joining tree of SMV-CP in our study. Hence, we stated that “there was no significant correlation between SMV population diversity and geographical isolation”. Account for the lacking of further population genetic analysis, our conclusion was thought to be rigorous, and we removed the conclusion in the revised manuscript (Line 123-124).

Reference:

Seo, J.K.; Ohshima, K.; Lee, H.G.; Son, M.; Choi, H.S.; Lee, S.H.; Sohn, S.H.; Kim, K.H. Molecular variability and genetic structure of the population of Soybean mosaic virus based on the analysis of complete genome sequences. Virology 2009, 393(1), 91-103.

Gao, L.; Wu, Y.; An, J.; Huang, W.; Liu, X.; Xue, Y.; Luan X.; Lin F.; Sun, L. Pathogenicity and genome-wide sequence analysis reveals relationships between soybean mosaic virus strains. Arch. Virol. 2022, 167(2), 517-529.

Comment 3: mice or rabbit. I think you used rabbit for PAb? (Line 149)

Answer: Yes, two female New Zealand White rabbits (two months’ old) were used for the development of PAb (Line 448-449, section 4, Materials and Methods), we have corrected it.

Comment 4: than 50%?. It is not making sense (Line 209)

Answer: Considering the accuracy and simplicity of the data description, the sentence “Four days after fusion, hybridoma cell growth well, and the cloning rate was higher than 50% with a small amount of cell debris.” have been removed in the revised manuscript.

Comment 5: remove highlighted sentence, it is antagonistic to previous and following sentence (Line 345-348)

Answer: Highlighted sentence has been removed.

Comment 6: Freund’s incomplete? (Line 445)

Answer: Yes, Freund's complete adjuvant is used for primary immunizations, while Freund's incomplete adjuvant is used for subsequent immunizations as Pertmer et al. (1996) descripted. 

Reference: Pertmer, T.M.; Roberts, T.R.; Haynes, J.R. Influenza virus nucleoprotein-specific immunoglobulin G subclass and cytokine responses elicited by DNA vaccination are dependent on the route of vector DNA delivery. J. Virol. 1996, 70(9), 6119-6125.

Comment 7: Did you confirmed the plants samples detected positive in immuno-strips with PCR and sequencing?

Answer: The specific primer pair used for RT-PCR has been confirmed according to the size of the PCR product with gel electrophoresis detection results. Hence, we have only conducted RT-PCR, without sequencing, to confirmed the plants samples detected positive in immuno-strips.

Reviewer 2 Report

General comments.

In this study entitled “Development of comprehensive serological techniques for sensitive, quantitative and rapid detection of soybean mosaic virus” authors report the production of monoclonal and polyclonal antibody production against SMV-CP. For this purpose, they have used the conserved region of CP protein. However, this approach is very conventional, and this study have lack novelty, except that it is developed against a particular virus.

Specific comments:

Line 92: Why SMV strains SC3, SC7, SC15 and SC18 were specifically selected for this study? What is the need to purify the above-mentioned strains, since many of the SMV sequences are already available?

Line 140: “Escherichia coli expression strain Arctic Express” Which Arctic express? Arctic express BL21 (DE3) or Arctic express BL21 (DE3) pLYS. In the material and methods section cite the following articles that describes expression, and purification viral protein using Arctic express strains,

Gnanasekaran P, Gupta N, Ponnusamy K, Chakraborty S (2021) Geminivirus Betasatellite-Encoded betaC1 Protein Exhibits Novel ATP Hydrolysis Activity That Influences Its DNA-Binding Activity and Viral Pathogenesis. J Virol 95 (17):e0047521. doi:10.1128/JVI.00475-21

Gnanasekaran P, Ponnusamy K, Chakraborty S (2019) A geminivirus betasatellite encoded betaC1 protein interacts with PsbP and subverts PsbP-mediated antiviral defence in plants. Mol Plant Pathol 20 (7):943-960. doi:10.1111/mpp.12804

 Line 146: Whether the purified proteins were confirmed by western blotting or by MALDI-TOF or Mass spec?

Figure 5C: Why this ELISA was not performed in the signal plate. The color development would vary too much in different plates. Moreover, the color developed with the chlorogenic substrate seems very light.

Ideally, while using an antiserum or polyclonal antibody, one should include pre immunized mice serum as negative control antibody. This is lacking in this study.

Line 485: Cite the following article that clearly explains the hybridoma procedure,

Prabu G, Iyer YS, Shankarkumar U, Ghosh K, Nachiappan V (2009) Monoclonal antibody produced against calf thymus histone. Hybridoma (Larchmt) 28 (4):277-280. doi:10.1089/hyb.2009.0005

Discussion: This section is poor. Try to explain why the developed method is timely important and interesting.

Author Response

Comment 1: In this study entitled “Development of comprehensive serological techniques for sensitive, quantitative and rapid detection of soybean mosaic virus” authors report the production of monoclonal and polyclonal antibody production against SMV-CP. For this purpose, they have used the conserved region of CP protein. However, this approach is very conventional, and this study have lack novelty, except that it is developed against a particular virus.

Answer: We have addressed all the issues and revised the manuscript carefully. The changes made in the manuscript are tracked and the itemized list of responses to the comments is as following.

Specific comments:

Comment 2: Line 92: Why SMV strains SC3, SC7, SC15 and SC18 were specifically selected for this study? What is the need to purify the above-mentioned strains, since many of the SMV sequences are already available?

Answer: As described in ‘Introduction’ that “The four of the most prevalent SMV strains, SC3, SC7, SC15 and SC18 have been reported widely distributed in the three major soybean-producing regions in China including Northeast China, Huang-Huai Valleys, and Southern China” (Line 43-46). Hence, SMV strains SC3, SC7, SC15 and SC18 were specifically selected for this study. As described in ‘Results 2.1 section’ that “To obtain purified isolate of SMV strains SC3, SC7, SC15 and SC18, biological purification and identification of the four SMV strains were carried out” (Line 91-92). In order to obtain purified isolates and to confirm the known SMV-cp sequences for sequence alignments, phylogenetic analysis and SMV-CP protein expression, biological purification and identification of SC3, SC7, SC15 and SC18 were carried out. 

Comment 3: Line 140: “Escherichia coli expression strain Arctic Express” Which Arctic express? Arctic express BL21 (DE3) or Arctic express BL21 (DE3) pLYS. In the material and methods section cite the following articles that describes expression, and purification viral protein using Arctic express strains,

Gnanasekaran P, Gupta N, Ponnusamy K, Chakraborty S (2021) Geminivirus Betasatellite-Encoded betaC1 Protein Exhibits Novel ATP Hydrolysis Activity That Influences Its DNA-Binding Activity and Viral Pathogenesis. J Virol 95 (17):e0047521. doi:10.1128/JVI.00475-21

Gnanasekaran P, Ponnusamy K, Chakraborty S (2019) A geminivirus betasatellite encoded betaC1 protein interacts with PsbP and subverts PsbP-mediated antiviral defence in plants. Mol Plant Pathol 20 (7):943-960. doi:10.1111/mpp.12804

Answer: In the ‘Materials and Methods’ section (Line 421-422), we described that “then transferred into the Escherichia coli expression strain ArcticExpress (DE3)”. Following carefully cross-check, we confirmed that the Escherichia coli expression strain should be revised as Arctic express BL21 (DE3) rather than Arctic express BL21 (DE3) pLYS. The two recommend articles that describes expression, and purification viral protein using Arctic express strains have been cited in the ‘material and methods’ section.

Comment 4: Line 146: Whether the purified proteins were confirmed by western blotting or by MALDI-TOF or Mass spec?

Answer: The purified proteins were confirmed by SDS-PAGE (Figure 3E, Figure S2), rather than western blotting or by MALDI-TOF or Mass spec.

Comment 5: Figure 5C: Why this ELISA was not performed in the signal plate.

Answer: The ELISA in Figure 5C was performed with three repeats requiring more than 180 ELISA wells, so it was not performed in the signal plate.

 Comment 6: The color development would vary too much in different plates.

Answer: As shown in the standard curve (Figure 5A), the OD450 value and the concentration of the SMV-CP protein were positively correlated. The color of the ELISA wells is known to be positively correlated with their OD450 value. Hence, the differences of the color for negative control, SC3, SC7, SC15 and SC18 were mainly due to the differences in the concentrations of the SMV-CP protein present in those samples, which has been verified in Figure 5D.

 Comment 7: Moreover, the color developed with the chlorogenic substrate seems very light.

Answer: Though the color developed with the chlorogenic substrate seems light, while their OD450 value were approximately 1.0-2.0 which meet the experimental requirements of ELISA.

 Comment 8: Ideally, while using an antiserum or polyclonal antibody, one should include preimmunized mice serum as negative control antibody. This is lacking in this study.

Answer: As a defect of our experimental design, preimmunized mice serum has not been collected as negative control antibody, and we will consider about in our future research.

Comment 9: Line 485: Cite the following article that clearly explains the hybridoma procedure,

Prabu G, Iyer YS, Shankarkumar U, Ghosh K, Nachiappan V (2009) Monoclonal antibody produced against calf thymus histone. Hybridoma (Larchmt) 28 (4):277-280. doi:10.1089/hyb.2009.0005

Answer: We have read through the article, and cited it to explains the hybridoma procedure Line 509.

Comment 10: Discussion: This section is poor. Try to explain why the developed method is timely, important and interesting.

Answer: Thanks for your suggestion, the “Discussion” has been revised.

Round 2

Reviewer 2 Report

Authors have addressed all my previous comments/suggestion. I recommend to accept this manuscript for publication.